

# The DNA methylation landscape of five pediatric-tumor types

Alyssa C. Parker, Badí I. Quinteros and Stephen R. Piccolo

Department of Biology, Brigham Young University, Provo, Utah, United States

## ABSTRACT

Fewer DNA mutations have been identified in pediatric tumors than in adult tumors, suggesting that alternative tumorigenic mechanisms, including aberrant DNA methylation, may play a prominent role. In one epigenetic process of regulating gene expression, methyl groups are attached at the 5-carbon of the cytosine ring, leading to 5-methylcytosine (5mC). In somatic cells, 5mC occurs mostly in CpG islands, which are often within promoter regions. In Wilms tumors and acute myeloid leukemias, increased levels of epigenetic silencing have been associated with worse patient outcomes. However, to date, researchers have studied methylation primarily in adult tumors and for specific genes—but not on a pan-pediatric cancer scale. We addressed these gaps first by aggregating methylation data from 309 noncancerous samples, establishing baseline expectations for each probe and gene. Even though these samples represent diverse, noncancerous tissue types and population ancestral groups, methylation levels were consistent for most genes. Second, we compared tumor methylation levels against the baseline values for 489 pediatric tumors representing five cancer types: Wilms tumors, clear cell sarcomas of the kidney, rhabdoid tumors, neuroblastomas, and osteosarcomas. Tumor hypomethylation was more common than hypermethylation, and as many as 41.7% of genes were hypomethylated in a given tumor, compared to a maximum of 34.2% for hypermethylated genes. However, in known oncogenes, hypermethylation was more than twice as common as in other genes. We identified 139 probes (31 genes) that were differentially methylated between at least one tumor type and baseline levels, and 32 genes that were differentially methylated across the pediatric tumor types. We evaluated whether genomic events and aberrant methylation were mutually exclusive but did not find evidence of this phenomenon.

## INTRODUCTION

Pediatric tumors are the leading cause of disease-related death for children in developed countries (*Filbin & Monje, 2019*), and those who survive pediatric cancer often experience adverse health challenges later in life (*Oeffinger et al., 2000*). Many mutations and structural variants have been associated with adult forms of cancer (*Stratton, Campbell & Futreal, 2009*); however, significantly fewer genomic abnormalities have been identified in pediatric tumors (*Yiu & Li, 2015*). The mutation rate in pediatric tumors is 14 times less than the mutation rate in adult tumors, implying that different mechanisms may be

Corresponding author
Stephen R. Piccolo,
stephen_piccolo@byu.edu

involved in pediatric-cancer development than in adult cancers (*Gröbner et al., 2018*). Of the mutations that have been identified in pediatric tumors, many are associated with epigenetic regulation (*Filbin & Monje, 2019*). In many pediatric tumors, molecular profiling does not identify genomic abnormalities but does show abnormal DNA methylation patterns (*Mack et al., 2014*; *Bayliss et al., 2016*), suggesting that DNA methylation may play a critical role in tumorigenesis in these cases.

Gene-expression levels in cancer cells often vary from those in normal cells (*Ehrlich, 2009*). Such abnormalities alter cellular environments and manipulate cellular processes, leading to increased survival, rapid proliferation, and metastasis (*Hanahan & Weinberg, 2011*). Oncogenes are one type of gene that contribute to the development of these abnormal features. These genes are often expressed at higher levels in cancer cells than in normal cells (*Croce, 2008*). In contrast, tumor suppressor genes counteract cellular changes that lead to cancer. These genes are often expressed at lower levels in tumor cells than in normal cells (*Macleod, 2000*), potentially leading to rapid cellular proliferation. Methylation of the promoter region is often negatively correlated with gene expression levels, suggesting that DNA methylation plays a role in regulating gene expression (*Esteller, 2007*; *Spainhour et al., 2019*). However, relatively little is known about global methylation patterns in pediatric tumors, the interplay between methylation events and mutations in pediatric tumors, or how these observations may differ between categories of known cancer genes (oncogenes or tumor suppressors) and other genes. Prior studies have focused on cancer cell lines, a single tumor type at a time, or adult cancers (*Dunwell et al., 2009*; *Liu et al., 2020*; *Shi et al., 2020*).

Computational models have been developed to identify differentially expressed genes across large sets of methylation data (*Saghafinia et al., 2018*; *Shi et al., 2020*). Applications of these methods have found several genes with highly variant methylation in adult tumors (*Saghafinia et al., 2018*). Many of the genes that exhibited highly variant methylation in tumors were not previously associated with cancer, and genomic aberrations in these genes were not characteristic of tumors. A small-scale study of Wilms tumors (a common pediatric cancer) showed similar results (*Saghafinia et al., 2018*). It has been shown that several distinct types of cancer, including endometrioid adenocarcinomas and glioblastomas, share many differentially methylated regions, suggesting that these epigenetic markers may be a universal feature of cancer (*Karlow et al., 2021*).

While these findings hint that aberrant methylation patterns may also characterize pediatric cancer, most pediatric cancers have not been analyzed for DNA methylation patterns. One study investigated genomic, transcriptomic, and epigenomic patterns in acute myeloid leukemia and identified dozens of hypermethylated genes and age-specific patterns (*Bolouri et al., 2018*). Additionally, structural variations were found to be more common than single nucleotide polymorphisms. A separate analysis of acute lymphoblastic leukemia identified a number of genes on chromosome 3, including *PPP2R3A*, *THRB* and *FBLN2*, which were frequently hypermethylated (*Dunwell et al., 2009*).

Little work has been done to specifically investigate aberrant methylation of oncogenes and tumor suppressor genes in cancer. One study about endometrial cancer identified seven oncogenes that were hypomethylated and upregulated and twelve tumor suppressor genes that were hypermethylated and downregulated (*Liu et al., 2020*), suggesting that changes in DNA methylation impact gene expression and may target oncogenes and tumor suppressor genes.

We address these gaps by analyzing methylation data for five types of pediatric cancers: Wilms tumors, clear cell sarcomas of the kidney, rhabdoid tumors, neuroblastomas, and osteosarcomas. As a baseline reference, we compare the tumor data against methylation levels from fetuses and children representing normal conditions for diverse cell types and human populations. Furthermore, we compare these cancer types against each other. Because methylation events tend to be gene specific (*Issa & Kantarjian, 2009*), we evaluate gene-level patterns in addition to probe-level patterns. We also consider global methylation patterns. For many of the tumors, we identify genomic alterations— single-nucleotide variants, small indels, and structural variants—in the tumors and evaluate whether these somatic mutations exhibited gene-specific mutual exclusivity with hypo- or hypermethylation events. In addition, we evaluate the consistency of these findings for oncogenes and tumor suppressor genes.

## RESULTS

Our goal was to evaluate DNA methylation patterns for pediatric-tumor cells and normal cells. We used publicly available data to characterize five types of pediatric cancers as well as baseline methylation levels for normal cells. First, we evaluated the consistency of methylation levels representing non-cancerous states in fetuses and children. Second, we compared tumor methylation levels against the normal values and identified genes and tumor samples that exhibited patterns of hypomethylation or hypermethylation. Next, under the assumption that tumors with aberrant methylation would be subject to evolutionary constraints that are redundant with those resulting from somatic mutations, we evaluated whether these two event types were mutually exclusive in a given tumor. Finally, we examined these patterns within known oncogenes and tumor-suppressor genes.

### Consistency of methylation levels in normal samples and in pediatric tumors

To aid in understanding how methylation levels change in cancer cells, we first characterized baseline methylation levels for individual genes. We obtained Illumina Infinium 450 K data for four normal datasets. We used data from diverse cell types and human populations with the goal of identifying methylation patterns that broadly represent baseline methylation states for healthy children. The data were from chorionic villi, kidney, spinal cord, brain, muscle, nasal epithelial, and blood cells and included data for a total of 309 patients of North American ($n = 94$), African American ($n = 36$), and Australian ancestry ($n = 179$). In total, 158 (51.1%) of the patients were female. After normalizing the microarray data and excluding unreliable probes and probes associated with regions that are distant from transcription start sites (see Methods), data for 88,178

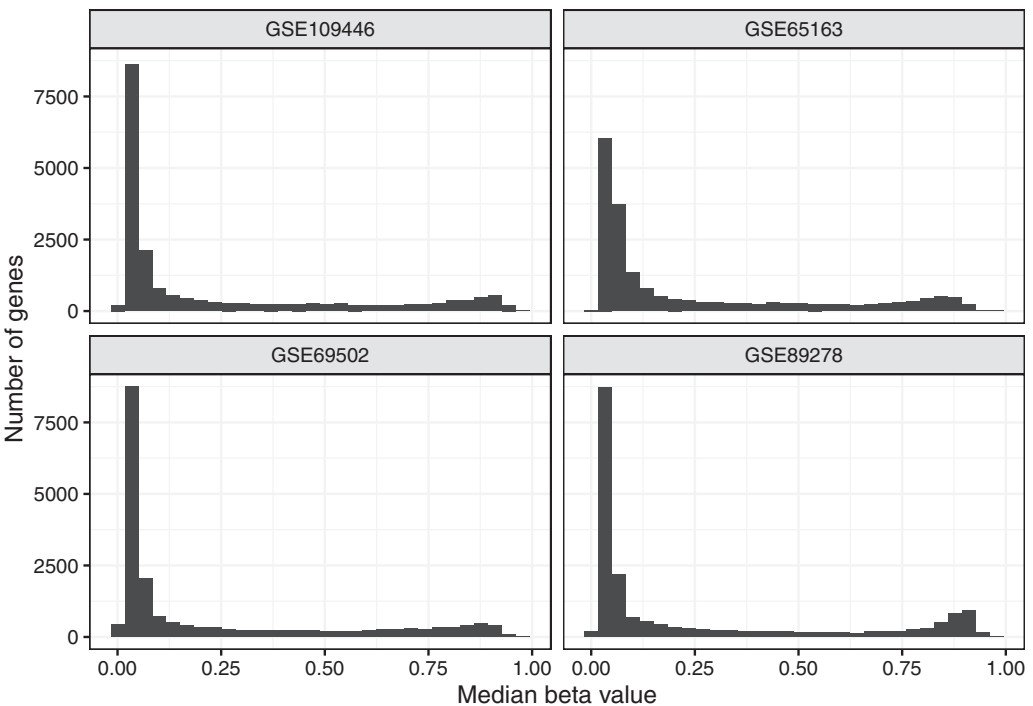

**Figure 1 Median beta values per gene in normal datasets.** We calculated the median beta value per gene across all patients in a given normal dataset.

probes and 19,184 genes remained. For most analyses, we used *beta* values, which fall between 0 and 1 and summarize the relationship between methylated and unmethylated signals in the data. Figure 1 and Figure S1 show the median beta values across all samples per normal dataset, either at the probe level or gene level.

Because DNA methylation contributes to regulating gene expression, we expected most genes to exhibit a consistent baseline methylation range. We anticipated that many genes (including some but not all tumor suppressor genes) would have relatively low methylation levels because those genes often regulate cellular division, growth, and other proliferation activities. We anticipated that other genes (including some but not all oncogenes) would have relatively high methylation levels because proper cellular functioning often requires that oncogenes remain inactive. We expected to see some variance across samples because we included data from multiple cell types and because methylation levels change as cells respond to internal and external cues. We also anticipated that some genes would deviate from these patterns, perhaps in part because they are regulated by mechanisms other than DNA methylation. Based on a preliminary inspection of the data, we identified thresholds for categorizing genes based on the magnitude and variance of methylation levels. We considered genes with a median value less than 0.2 across all samples in a given dataset to be methylated at "low" levels, genes with a median greater than 0.6 to be methylated at "high" methylation levels and the remaining genes to have "medium" methylation levels. We categorized genes with a coefficient of variation (CV) less than 0.5 in a given dataset as having "low" variance and the remaining genes as having "high" variance. Across all datasets representing normal

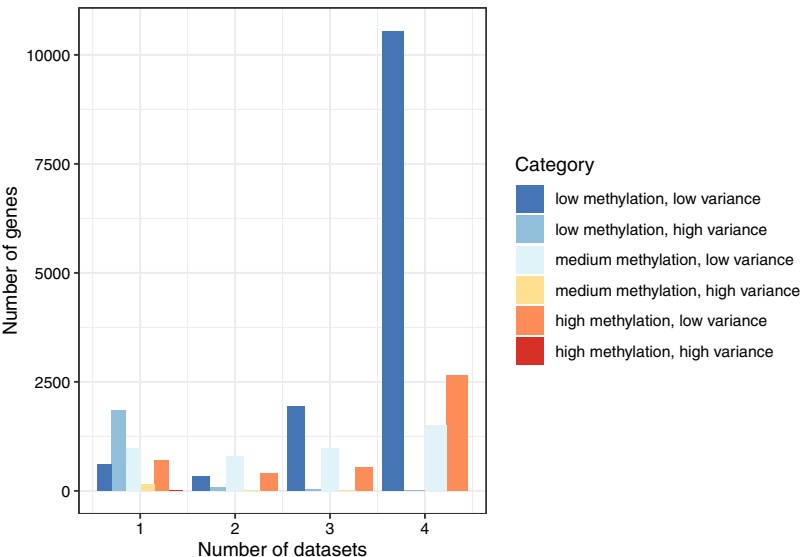

**Figure 2 Consistency of DNA methylation levels and variances in normal cells.** We assigned each gene to a category that indicated whether it was methylated at low, medium, or high levels and whether it had low or high variance across samples in a given dataset. We calculated the number of genes per combination and the number of normal datasets that shared the same category.

tissues, 67.2% of genes exhibited low methylation levels, while 19.7% were methylated at high levels, suggesting that most genes appear not to be subject to strong expression constraints as a result of DNA methylation. Nearly all genes (91.7%) exhibited low variance across the normal datasets Figs. S2 and S3.

By combining these two ways of categorizing the genes, we found that the most common combination across all datasets was low methylation/low variance (58.9–66.9% per normal dataset). The least common combination was high methylation/high variance, which occurred only eight times total (Table S1). To summarize these findings, we classified each gene as *consistent* (same category in all four datasets), *semiconsistent* (same category in three datasets), or *inconsistent* (same category in two or fewer datasets). Of the 19,184 genes, 14,706 (76.7%) were consistent, 3,481 (18.1%) were semiconsistent, and 997 (5.2%) were inconsistent (Fig. 2). These findings demonstrate that even though the normal datasets differed based on cell type and population ancestral groups, gene-level methylation levels were largely consistent under normal conditions. However, some patterns were specific to a given dataset. Of the 2,106 genes that were highly variant for at least one dataset, 991 (47.1%) genes were highly variant when calculated across all the normal datasets, showing that genes with high variance in one dataset often did not exhibit high variance more generally.

Using Gene Ontology terms (*Consortium, 2019*), we performed a gene-set analysis based on genes associated with the 100 lowest-variance probes or 100 highest-variance probes across the normal samples; however, in both cases, no gene set showed a significant enrichment for these probes.

**Table 1 Summary of changes in methylation level/variance categories between normal and cancer datasets.** We assigned each gene to a category that indicated whether it was methylated at low, medium, or high levels and whether it had low or high variance across samples in a given dataset. The table shows the total number of genes in each category for the normal datasets and the number (and percentage) of genes that changed from one category to another in the tumor datasets.

| Normal category | Tumor category | Total # genes (normal) | # genes changed | % genes changed |
|---|---|---|---|---|
| Low methylation/low variance | High methylation/low variance | 12,826 | 86 | 0.67 |
| Low methylation/low variance | Low methylation/high variance | 12,826 | 1,568 | 12.2 |
| Low methylation/low variance | Medium methylation/high variance | 12,826 | 88 | 0.69 |
| Low methylation/low variance | Medium methylation/low variance | 12,826 | 382 | 2.98 |
| Low methylation/high variance | High methylation/low variance | 118 | 4 | 3.39 |
| Low methylation/high variance | Low methylation/low variance | 118 | 43 | 36.4 |
| Low methylation/high variance | Medium methylation/high variance | 118 | 4 | 3.39 |
| Low methylation/high variance | Medium methylation/low variance | 118 | 3 | 2.54 |
| Medium methylation/low variance | High methylation/high variance | 3,274 | 1 | 0.03 |
| Medium methylation/low variance | High methylation/low variance | 3,274 | 743 | 22.7 |
| Medium methylation/low variance | Low methylation/high variance | 3,274 | 277 | 8.5 |
| Medium methylation/low variance | Low methylation/low variance | 3,274 | 244 | 7.5 |
| Medium methylation/low variance | Medium methylation/high variance | 3,274 | 150 | 4.6 |
| Medium methylation/high variance | High methylation/low variance | 10 | 4 | 40.0 |
| Medium methylation/high variance | Low methylation/high variance | 10 | 1 | 10.0 |
| Medium methylation/high variance | Medium methylation/low variance | 10 | 2 | 20.0 |
| High methylation/low variance | High methylation/high variance | 3,576 | 1 | 0.03 |
| High methylation/low variance | Low methylation/high variance | 3,576 | 27 | 0.76 |
| High methylation/low variance | Low methylation/low variance | 3,576 | 4 | 0.11 |
| High methylation/low variance | Medium methylation/high variance | 3,576 | 136 | 3.8 |
| High methylation/low variance | Medium methylation/low variance | 3,576 | 583 | 16.3 |

## Tumor methylation relative to normal levels

We obtained methylation data for 121 Wilms tumors, 11 clear cell sarcomas of the kidney (CCSK), 68 rhabdoid tumors, 203 neuroblastomas, and 86 osteosarcomas. Across all tumor types, 221 patients (45.2%) were female. All CCSK patients were male.

Under the hypothesis that tumorigenesis alters local and global methylation patterns, we evaluated the extent to which methylation levels and variances differed for a given gene between normal and tumor conditions. For each gene, we compared the most common median/variance category across the four normal datasets against the most common category across the five cancer datasets. Of the 12,826 genes that were categorized as low methylation/low variance for the normal datasets, 2,124 (16.6%) changed categories, most frequently to low methylation/high variance ($n = 1,568$) or to medium methylation/ low variance ($n = 382$) (Table 1). The 118 genes in the low methylation/high variance category were most likely to change categories, with 43 genes (36.4%) changing to low methylation/low variance. Of the 4,351 genes that changed categories, methylation levels increased (from low to medium or medium to high) for 1,989 (45.7%) genes. Overall, of the genes that changed categories, 2,248 (51.7%) increased from low to high variance.

These changes confirm that the factors that keep methylation levels stable under normal conditions often become dysregulated in tumors. Altered expression of these genes may play a role in tumor development or may lead to downstream effects that affect tumor development. For example, if methylation levels of an oncogene are decreased, higher expression of the gene may result, leading to increased cellular growth, proliferation, or survival (*Hanahan & Weinberg, 2011*). On the other hand, increased methylation levels of a tumor suppressor gene could cause lower gene expression and prevent it from performing regulatory functions (*Esteller, 2007*).

To identify factors that may influence pediatric tumorigenesis, we compared tumor methylation levels for each microarray probe against the respective normal values on a per-cancer basis. We considered probes with a False Discovery Rate (FDR) (*Benjamini & Hochberg, 1995*) smaller than 0.05 and an absolute, log2-transformed, fold-change of 2 or greater to be significantly different. Of the 10,446 probes that remained after prior filtering steps, 139 were differentially methylated (Table S2). For 120 (86.3%) of the probes, the differences were significant for only a single tumor type. We applied the same methodology at the gene level and observed similar results. Of the 2,259 genes that remained after filtering, 32 showed a significant difference for at least one tumor type (Table S3, Fig. 3), and these differences were specific to a single tumor type for 27 of the genes. *TSTD1* was the only gene methylated at significantly different (higher) levels in three tumor types: clear cell sarcomas of the kidney, osteosarcomas, and rhabdoid tumors (Fig. S4). We performed a gene-set analysis for the significant probes from each tumor type. After adjusting for multiple tests, no Gene Ontology term was statistically significant.

To characterize differences in methylation across the five cancer types, independent of normal methylation levels, we generated linear models that contrasted each pair of tumor types while adjusting for sex as a covariate. We focused on probes and genes that had an FDR value below 0.05 and exhibited an absolute, log-2 transformed fold change between any tumor-type pair of 2 or greater. Of the 139 probes that differed across the tumor types, 78 (56.1%) were also significantly different between tumors and normal tissues. Methylation levels for 31 genes differed significantly across the tumor types (Table S4). The largest absolute difference between tumor types was a 2.9-fold increase of *SPRY2* methylation in osteosarcomas relative to clear cell sarcomas of the kidney. Methylation levels for this gene were more than twice as high in osteosarcomas relative to Wilms tumors, which also affect the kidneys Fig. 4. The Spry2 protein has been associated with lower expression in (adult) renal cell carcinomas than in adjacent normal cell and correlated with clinicopathologic findings such as tumor size and poor survival (*Li et al., 2013*); however, these results may conflict with our observation of relatively low methylation levels in kidney tumors, perhaps due to differences between pediatric and adult cases. A functional analysis has shown that *SPRY2* helps to inhibit the MAPK/ERK pathway, which influences cellular proliferation (*Tsavachidou et al., 2004*).

For the 20 genes that varied most across the tumor types, we plotted the methylation values relative to normal levels for each patient (Fig. 5). Although in some cases the same genes were expressed aberrantly in multiple tumor types, the overall methylation patterns differed considerably across the tumor types.

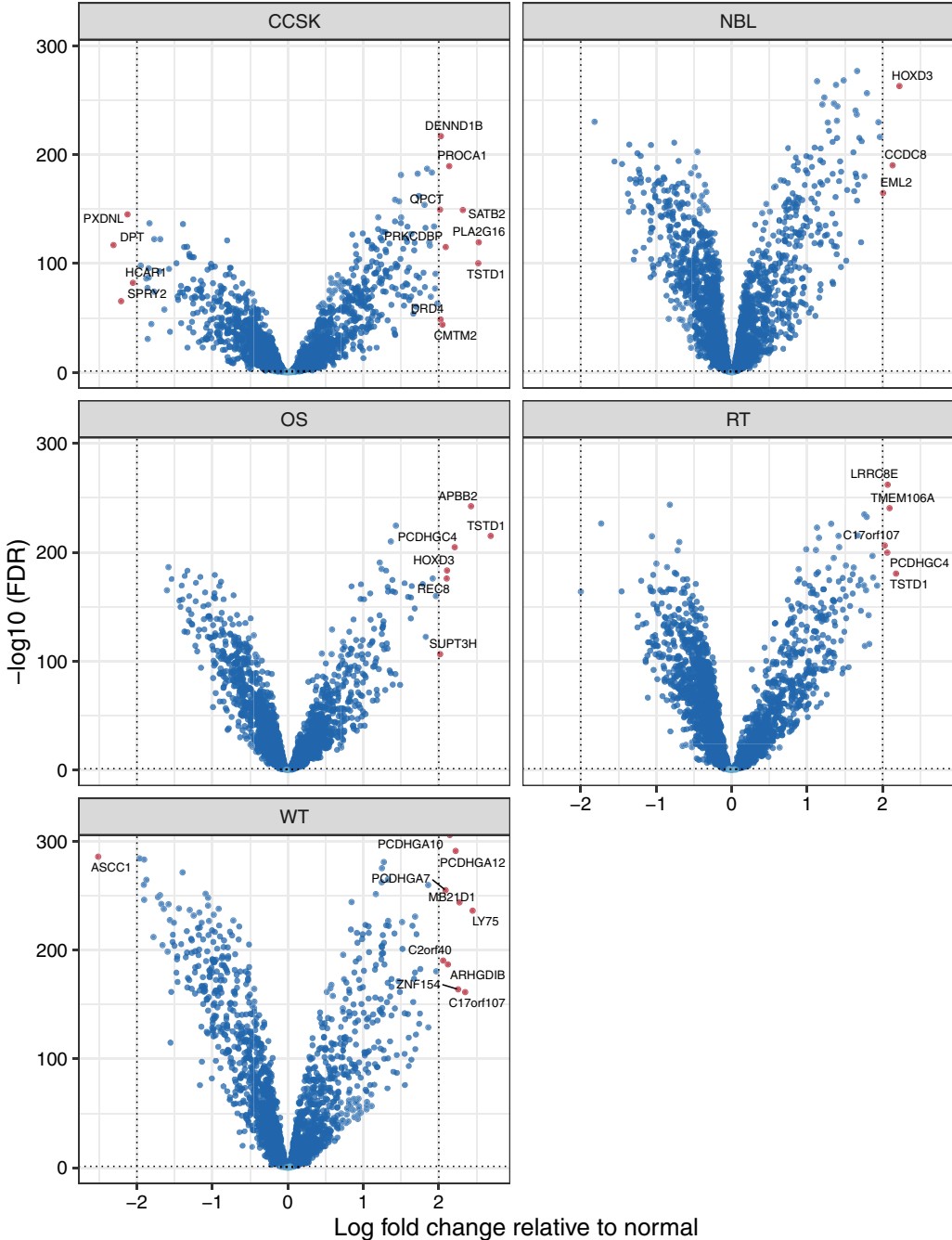

**Figure 3 Volcano plots showing differentially methylated genes for each tumor type.** For each tumor type, we compared methylation levels at the gene level between tumors and the normal samples. Genes showing significantly different methylation levels between tumor and normal conditions are highlighted.

To investigate global methylation patterns, we calculated gene-level z-scores for each tumor using the normal data as a reference. In cases where a tumor's methylation value was more than three standard deviations higher than the mean normal value for a particular gene, we classified that gene as being hypermethylated in that tumor. In cases

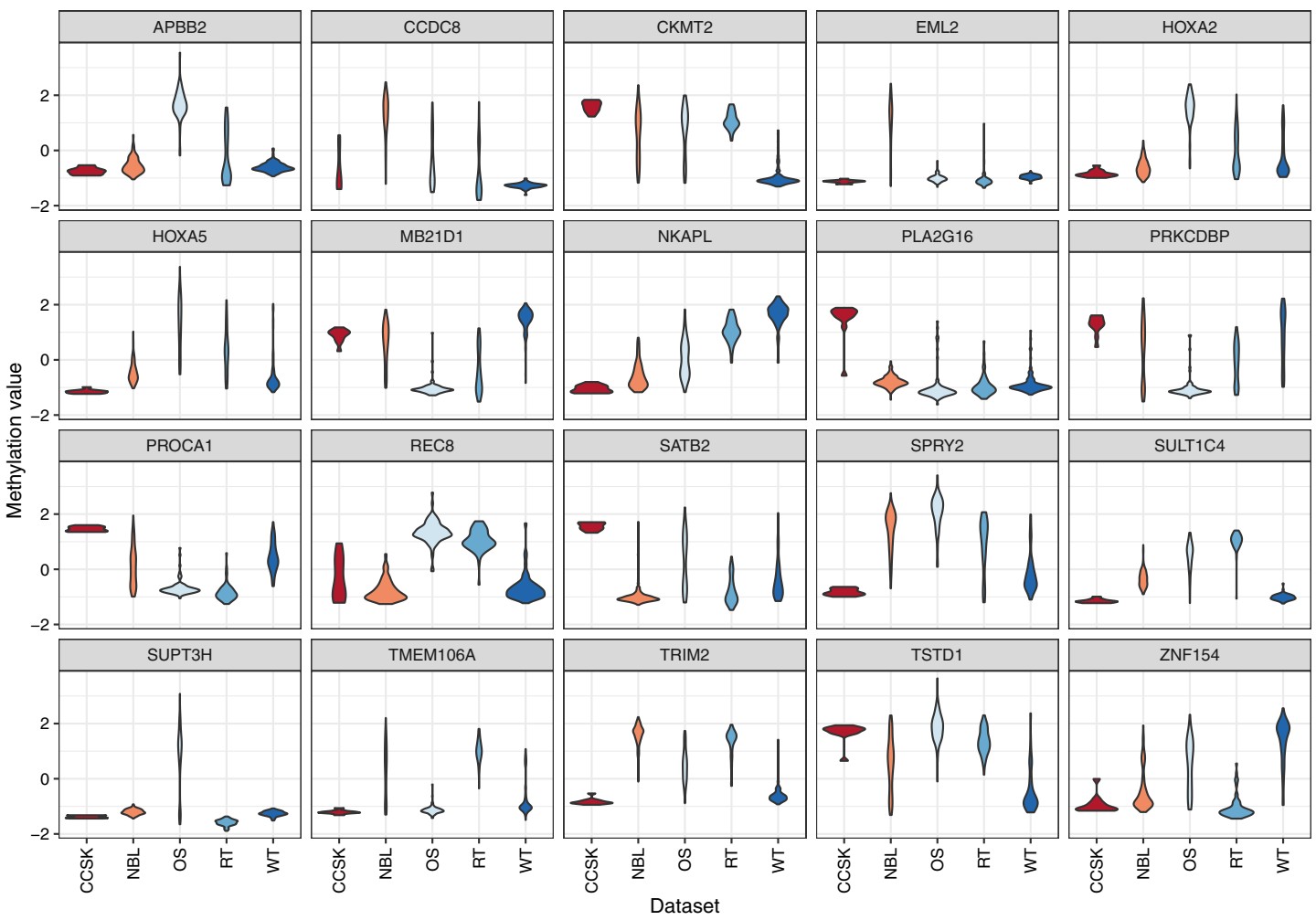

**Figure 4** **Methylation levels for genes that differed significantly across the tumor types.** Comparisons across tumor types identified 31 genes for which the methylation levels differed significantly. These violin plots show the range and density of the methylation values for 20 of these genes.

where a tumor's methylation value was more than three standard deviations lower than the mean normal value for a particular gene, we classified that gene as being hypomethylated in that tumor. Across all tumor types, the proportion of hypomethylated genes was higher than the proportion of hypermethylated genes with some tumors showing hypomethylation for more than 40% of genes (Fig. 6).

## Cancer mutation data analysis

Although they occur less frequently in pediatric tumors than in adult tumors, somatic mutations often contribute to pediatric tumorigenesis (*Yiu & Li, 2015*; *Gröbner et al., 2018*). To evaluate the frequency of and interplay between somatic mutations and methylation events, we examined pediatric tumors for which both data types were available. Mutation data were available for somatic single nucleotide variants (SNVs), insertions/deletions (indels), and RNA fusions. We used the RNA fusion data as indicators of structural DNA variants. Copy-number data were available for only a small number of

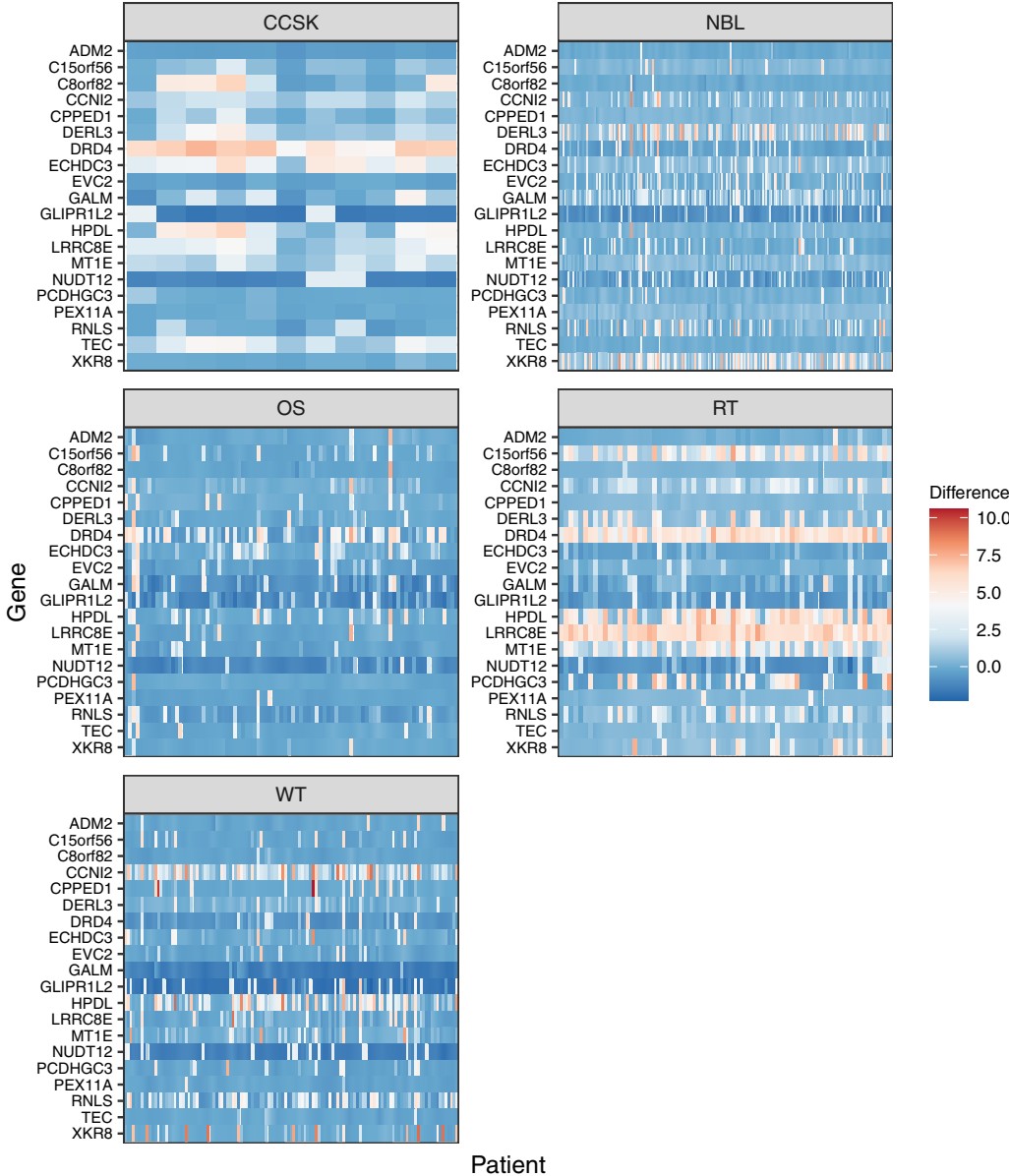

**Figure 5 Gene-level DNA methylation changes for high-variance genes.** Rows in these heatmaps indicate methylation levels, relative to the normal data, for the 20 genes with the largest variance across the tumor types. Columns represent individual tumors.

tumors, so we did not include this data type in the analysis. Methylation, SNV, indel, and RNA fusion data were available for Wilms tumors ($n = 41$), neuroblastomas ($n = 65$), and osteosarcomas ($n = 66$) but not for the other tumor types. For a given tumor, we considered genes with at least one SNV, indel, or RNA fusion event to be "mutated." According to our methodology, aberrant methylation—*via* either hypomethylation or hypermethylation of a given gene—occurred much more frequently (in 37–49% of tumors) than mutations (2.8%). The highest percentage of aberrant methylation occurred in rhabdoid tumors (49.2%).

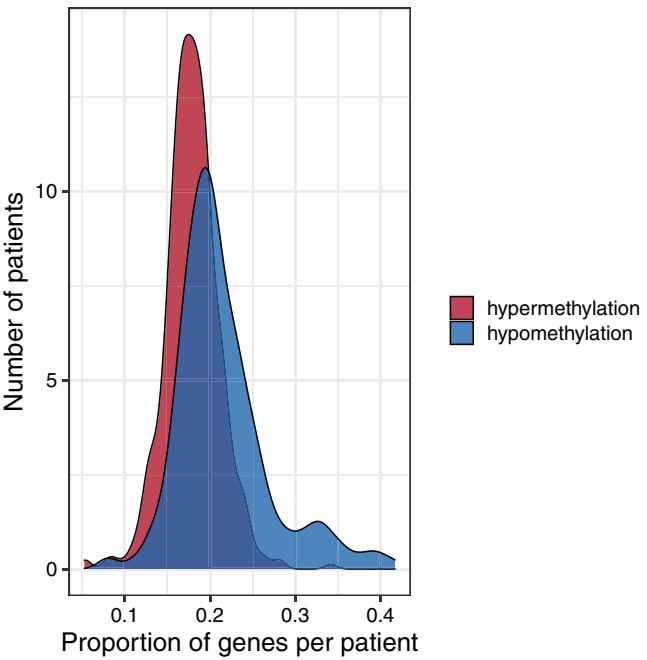

**Figure 6 Distributions of the proportion of hypermethylated or hypomethylated genes in a given tumor.** Using the normal data as a reference, we identified genes that were hypermethylated or hypomethylated in a given tumor. All five tumor types are represented. A relatively large number of hypomethylated genes in a given tumor was more common than a relatively large number of hypermethylated genes.

Aberrant methylation levels and damaging mutations can have similar downstream effects in tumors (*Bodily et al., 2020*). After one of these alteration types has occurred in a given gene, it may be less likely for cells with a second alteration in the same gene to gain an additional selective advantage. Thus, having both a mutation and aberrant methylation in one gene may be less likely than expected by random chance. This mutual-exclusivity hypothesis has been examined extensively for pairs of genes in which somatic mutations might occur across diverse types of cancers (*Kang et al., 2008*; *Ciriello et al., 2012*; *Szczurek & Beerenwinkel, 2014*; *Babur et al., 2015*). It has also been investigated for DNA methylation events, though to a lesser extent (*Ding et al., 2020*). Little is known about mutual exclusivity between methylation events and somatic mutations in pediatric tumors.

Treating each combination of tumor and gene as an independent observation, we examined whether DNA methylation events are mutually exclusive with somatic mutations. Because both of these event types are rare, we reduced the data to the 503 genes for which at least five mutation events and at least five aberrant methylation events had occurred across the tumor types. Mutations and aberrant methylation co-occurred in the same gene and the same tumor in 1.8% of cases. We used a permutation test to evaluate whether mutations and aberrant methylation in the same gene and tumor co-occurred less frequently than would be expected by random chance. Across all tumors and the 503 genes, we observed 4,482 co-occurrences, whereas the average number of co-occurrences in the permuted data was 4,440.5; this difference was not statistically significant ($p = 0.87$).

**Table 2 Aberrant methylation and mutation rates in oncogenes, tumor suppressor genes, and all other genes.** For three pediatric tumor types, we identified aberrant methylation events (either hypomethylation or hypermethylation) that had occurred in a given tumor and gene. For the same tumor/gene combinations, we identified somatic single-nucleotide variants, indels, and structural variants that had occurred. These numbers indicate overall rates of aberrant methylation or mutation across all tumors of a given type. Methylation rates and mutation rates were typically similar across all three gene categories, but mutation rates for oncogenes and tumor suppressor genes were always higher than for other genes.

| Tumor type | Aberration type | Oncogene | TSG | Other |
|---|---|---|---|---|
| Neuroblastoma | Hypermethylation | 0.348 | 0.256 | 0.176 |
| Neuroblastoma | Hypomethylation | 0.158 | 0.145 | 0.203 |
| Neuroblastoma | Mutation | 0.052 | 0.058 | 0.035 |
| Osteosarcoma | Hypermethylation | 0.402 | 0.199 | 0.179 |
| Osteosarcoma | Hypomethylation | 0.089 | 0.110 | 0.203 |
| Osteosarcoma | Mutation | 0.027 | 0.036 | 0.021 |
| Wilms tumor | Hypermethylation | 0.477 | 0.194 | 0.171 |
| Wilms tumor | Hypomethylation | 0.048 | 0.152 | 0.197 |
| Wilms tumor | Mutation | 0.011 | 0.012 | 0.006 |

## Oncogenes and tumor suppressor genes

Because we expected that oncogenes would often be expressed at relatively low levels and tumor suppressor genes at relatively high levels under normal conditions, we expected that oncogenes would have higher methylation levels than tumor suppressor genes on average. We identified 80 "tier 1" oncogenes and 142 "tier 1" tumor suppressor genes in the Cancer Genome Census (*Tate et al., 2019*); 9 and 14 of the genes in these respective categories remained after probe-level filtering (see Methods). We calculated the mean $M$ methylation value for each gene in the normal datasets and used a two-sample t-test to evaluate whether these mean values differed between the oncogenes and tumor suppressor genes. Under the hypothesis that oncogenes would be methylated at higher levels than tumor-suppressor genes, we used a one-sided test. The mean of the means for oncogenes was considerably higher than for tumor-suppressor genes (Fig. S5); however, this difference was not statistically significant ($p = 0.08$). Furthermore, none of the oncogenes or tumor-suppressor genes were among those that showed a statistically significant difference between tumors and normal samples or across tumor types.

In neuroblastomas and Wilms tumors, mutation rates for oncogenes and tumor suppressor genes were nearly twice the mutation rates for other genes (Table 2), which aligns with prior evidence that mutations in these genes are associated with tumorigenesis. For all three cancer types with mutation data available, hypermethylation occurred more frequently in oncogenes and tumor suppressors than in other genes. In osteosarcomas and Wilms tumors, hypermethylation rates were more than twice as high in oncogenes as in non-cancer genes. Hypomethylation was much less common in oncogenes and tumor suppressors than in non-cancer genes. These findings suggest that cancer genes are
commonly *dysregulated* in tumors, even though the direction of dysregulation did not necessarily agree with our expectations.

We performed a modified version of the permutation-based, mutual-exclusivity analysis in which we searched for co-occurrences of either (1) a mutation in an oncogene and aberrant hypomethylation of the same gene or (2) a mutation in a tumor suppressor gene and hypermethylation of the same gene. These event combinations were not mutually exclusive for oncogenes ($p = 0.59$) nor tumor suppressors ($p = 0.08$).

## DISCUSSION

Using publicly available data, we examined methylation patterns for five childhood tumor types. We summarized baseline methylation levels for healthy children, identified deviations from these baseline levels in tumors, and evaluated mutual exclusivity of methylation events and somatic mutations. Subsequently, we evaluated these patterns for oncogenes and tumor-suppressor genes specifically. In the 309 healthy samples we studied, DNA methylation levels were mostly consistent, suggesting that biological processes maintain this consistency in diverse tissue types and ancestral groups. In the 489 tumors we studied, hypermethylation or hypomethylation of promoter regions was a common feature, providing additional evidence that the tumor epigenome contributes to pediatric tumorigenesis. Hypomethylation was more common than hypermethylation. Hypermethylation of promoter regions has been associated with decreased gene expression (*Spainhour et al., 2019*)—tumors with frequent hypermethylation may result in broad silencing of proteins necessary for normal cellular function. Hypermethylation affecting multiple genes in the same tumor has been identified in adult thyroid neoplasms (*Keelawat et al., 2015*) and colorectal tumors (*Toyota et al., 1999*) and has been associated with multidrug resistance in cell cultures (*Segura-Pacheco et al., 2006*). Global hypomethylation has garnered more attention than multigene hypermethylation (*Feinberg & Tycko, 2004*).

Perhaps surprisingly, the data showed that hypermethylation was more common in oncogenes and tumor suppressor genes than in other genes, while hypomethylation was less common. This pattern might reflect that we focused on CpG islands, whereas hypomethylation often manifests itself more broadly across the genome.

Our mutual-exclusivity analysis did not provide evidence that somatic mutations and aberrant methylation are mutually exclusive from each other, whether for oncogenes and tumor-suppressor genes specifically or across all genes. However, we had access to a limited number of tumors ($n = 172$) for which both mutation data and methylation data were available; a larger-scale analysis is warranted. Furthermore, we evaluated single-nucleotide variants, indels, and structural variants because data were available for these mutation types. But large-scale amplifications and deletions occur regularly in tumors (*Shlien & Malkin, 2009*), so including copy-number data in this type of analysis would also be useful in future research.

In performing these analyses, we needed to choose arbitrary thresholds at times. For example, we specified three standard deviations above or below the mean in the normal data as a conservative threshold to indicate aberrant methylation. Despite this stringent threshold, we found that tumors were aberrantly methylated much more

frequently than they were mutated. The use of arbitrary thresholds in research is common and cannot be completely avoided, especially when discipline-specific precedents have yet to be specified by the research community. In this scenario, we have introduced a way of identifying aberrant methylation events. Future quantitative, experimental, and translational work will be necessary to improve our ability to determine when a particular gene in a given biological sample is methylated to an extent that alters that gene's behavior and may in turn have clinical relevance.

A better understanding of DNA methylation's role in pediatric tumors could shed light on mechanisms of tumorigenesis and eventually lead to insights about patient care and treatments. DNA methylation inhibitors have proven effective in some adult cancers, especially hematologic malignancies (*Issa & Kantarjian, 2009*) and may prove beneficial in pediatric cases. However, these therapies primarily target hypermethylation in a broad sense and thus may not be suitable for targeting specific genes (*Cheishvili, Boureau & Szyf, 2015*; *Roberti et al., 2019*). Furthermore, as we have shown, hypomethylation events may be more common than hypermethylation events in pediatric tumors. Little is understood about how to reverse hypomethylation *in vivo*; however, global-hypomethylation patterns may be useful as biomarkers for therapies for targeting genes that have been activated as a result of hypomethylation.

## METHODS

### Normal methylation data acquisition and processing

We downloaded datasets with DNA methylation levels for cohorts of normal patients to establish a baseline against which pediatric tumor methylation could be compared. We selected four datasets from Gene Expression Omnibus that used Illumina Infinium HumanMethylation450K arrays. These included data representing normal cells in healthy fetuses and children aged 18 or under for which raw (.idat) files were available. Illumina Infinium HumanMethylation450K is a microarray platform that detects DNA methylation at over 450,000 locations in the human genome. Beta values from these arrays indicate the ratio of methylated signals to unmethylated signals (*Dedeurwaerder et al., 2014*). Values closer to one indicate relatively high methylation levels; values closer to zero indicate relatively low methylation levels. *M-values* are logit transformed and are more suitable when applying statistical tests (*Du et al., 2010*).

Here we describe each dataset in our study and provide accession identifiers from Gene Expression Omnibus.

GSE69502 originated from a study of Canadian second trimester fetuses (*Price et al., 2016*). Tissue was collected from chorionic villi, kidney, spinal cord, brain, and muscle. The original study analyzed DNA methylation differences in fetuses with spina bifida and anencephaly compared to normal fetuses. We included data from the 65 normal fetuses.

GSE65163 originated from a study of nasal epithelial cells from African American children aged 10–12 (*Yang et al., 2017*). The original study analyzed DNA methylation differences in children with or without asthma. We included data from the 36 children who did not have asthma.

GSE109446 originated from a study of children aged 5–18 living in Cincinnati, Ohio (USA) (*Zhang et al., 2018*). Similar to GSE65163, nasal epithelial cells were used, and the goal of the original study was to understand methylation differences in children with or without asthma. We included data from the 29 children who did not have asthma.

GSE89278 originated from a study of Australian infant blood spot samples taken at birth (*van Dijk et al., 2016*). The goal of the original study was to understand methylation differences in children that had or had not been exposed to docosahexaenoic acid *via* a nutritional supplement taken orally by the mothers. We included data from the 179 infants who were given a placebo.

We chose these samples because they represented a variety of cell types and ancestral groups, which we hoped would make our findings more broadly generalizable than if our samples had been from a single, homogeneous population. Another factor in our decision was the age of the individuals. We searched for datasets representing relatively young patients to limit the possible confounding factor of methylation changes accumulating throughout life. In this process, we considered one additional normal dataset (GSE72556) (*Oelsner et al., 2017*); however, we found that the beta levels from this dataset were consistently different from the other normal datasets. This dataset originated from saliva samples, which often result in systematically different methylation levels than other types of samples (*Armstrong et al., 2014*; *Smith et al., 2015*; *Godderis et al., 2015*), perhaps due to external contaminants or different sample-collection procedures. As a result, we excluded this dataset from our analysis.

We processed the data using the minfi package (version 1.34.0) from R (version 4.0.2) (*R Core Team, 2020*). and the Bioconductor suite (*Aryee et al., 2014*; *Amezquita et al., 2020*). We followed a standard workflow to process the methylation array files (*Maksimovic, Phipson & Oshlack, 2016*). The steps in this workflow include preprocessing, ratio conversion, and beta value calculations. Because we were interested in methylation changes in the promoter regions of genes, we included only probes within 300 base pairs of transcription start sites. For this mapping, we used an annotation file created by *Price et al. (2013)* (see http://www.ncbi.nlm.nih.gov/geo/query/acc.cgi?acc=GPL16304). We repeated these steps for beta values and M-values. For the remaining 88,178 probes, we summarized the values at the gene level (using the annotation file by *Price et al. (2013)*) for each patient as the mean probe value for a given gene.

For each normal dataset, we wrote custom code to parse the series matrix files from Gene Expression Omnibus and identify the sex and tissue type of each sample.

## Tumor methylation data acquisition and processing

We obtained methylation data representing five tumor types from the Therapeutically Applicable Research to Generate Effective Treatments (TARGET) Data Matrix (retrieved August 24, 2020 from website, https://ocg.cancer.gov/programs/target/data-matrix). We downloaded the data from TARGET using the rvest package (version 0.3.6) (*Wickham, 2020*). The data for Wilms tumors, clear cell sarcomas of the kidney, neuroblastomas, and osteosarcomas were generated using the Illumina Infinium HumanMethylation450K platform, and the data for rhabdoid tumors were generated

using the Illumina Infinium MethylationEPIC platform. We limited our analysis to the genes that overlapped between the two platforms. We followed the same workflow that we had used to process the normal data, preprocessing the probe-level values and summarizing at the gene-level.

We limited the analysis to primary tumors. To identify primary status and patient sex, we downloaded clinical variables from the Genomic Data Commons portal (*Pugh et al., 2013*; *Grossman et al., 2016*; *Gadd et al., 2017*) by going to the project homepage (such as https://portal.gdc.cancer.gov/projects/TARGET-NBL), clicking on the Clinical download button, and downloading the tab-separated-value (TSV) files.

## Somatic-mutation data acquisition and processing

We obtained somatic-mutation data for TARGET patients *via* the Genomic Data Commons portal. On that portal, we first selected the Repository section. Under the Cases tab, we specified the TARGET program. We also specified Wilms tumor, neuroblastoma, and osteosarcoma as the tumor types of interest (data were unavailable for the other two tumor types). Under the Files tab, we selected "simple nucleotide variation" and "annotated somatic mutation" as the data category and type, and we indicated that we would use variants that had been called using Mutect2 (*Benjamin et al., 2019*). The data were stored in VCF format (version 4.2) (*Danecek et al., 2011*).

We wrote custom code to parse the mutation data for each patient. We included only mutations that had either (1) "HIGH" impact according to the variant annotations or (2) had "MODERATE" impact and were considered to be "deleterious" by SIFT (*Vaser et al., 2016*) or either "damaging" or "probably_damaging" according to Polyphen-2 (*Adzhubei, Jordan & Sunyaev, 2013*). We considered using the specified minor-allele-frequency (MAF) values for filtering, but those values were unavailable for a significant portion of the variants, so we focused on functional annotations.

## RNA-fusion data acquisition and processing

We downloaded data for RNA fusions *via* the Genomic Data Commons portal. Under the Cases tab, we applied the same filters that we used for obtaining the somatic-mutation data. Under the Files tab, we selected "structural variation" and "Transcript Fusion" as the data category and type. We specified "STAR-Fusion" (*Haas et al., 2019*) as the workflow type and "bedpe" as the data format. We wrote custom code to parse the RNA fusion data for each patient. The bedpe files describe RNA transcripts from each patient that had genetic information originating from two separate genes. We considered an RNA fusion to have a functional impact on both genes affected by the fusion.

## Additional filtering and data preprocessing

We used the resulting data to characterize methylation levels (low, medium, or high) and the amount of variance (low or high) for the samples from a given dataset and across datasets. Before performing statistical tests between normal and tumor samples or across tumor types, we performed additional probe-filtering steps with the goal of reducing the data to genes most likely to reflect the underlying biology. We identified 35,548 probes

that fell into the same methylation-level category across all samples (normal and tumor) and excluded these probes from our analysis.

To perform the above-described filtering based on tumor expression, we obtained RNA-Sequencing data from the Genomic Data Commons portal and excluded the few samples from normal tissues. We used Ensembl BioMart (*Kinsella et al., 2011*) to map between Ensembl identifiers in the gene-expression files and HUGO gene symbols in the methylation data. We limited these steps to protein-coding genes and excluded genes for which no symbol was available. *UGT2A1* and *TIMM23B* had ambiguous mappings (two Ensembl identifiers each), so we excluded those. We also excluded genes that had the same value for every patient. This resulted in 18,814 unique genes. A total of 10,686 methylation probes were significantly correlated with tumor gene expression; we retained data these probes.

Lastly, we identified probes that were methylated at significantly different levels between any pair of normal datasets (*e.g.*, GSE69502 *vs.* GSE65163 or GSE59502 *vs.* GSE109446). We applied the limma package (*Ritchie et al., 2015*) (version 3.50) to the M values and included sex as a covariate. We considered probes to be significantly different when the adjusted $p$-value was lower than 0.05, and the absolute log fold-change was 2 or greater. We excluded the 49 probes that met these criteria.

We used principal component analysis to assess high-level patterns across the datasets and visualized the results using a scatter plot of the first two principal components. Methylation samples from each dataset generally clustered tightly with other samples from the same dataset, demonstrating that batch effects were present (Fig. S6). Some of this variance was expected, given that the samples originated from different tissue types. However, to support cross-tissue comparisons, including between normal samples and tumor samples, we used a linear model to adjust for these differences. We applied batch correction to a combined dataset that included all of the normal datasets. We used the dataset identifier as the batch variable. For the beta values, we logit transformed the data before performing the batch correction and inverse logit transformed the data after performing the batch correction to ensure the beta values stayed between 0 and 1. We performed these transformations using functions from the gtools package (version 3.8.2) (*Warnes, Bolker & Lumley, 2020*). To perform the batch correction, we used the ComBat function from the sva package (version 3.36.0) (*Leek et al., 2012*). A subsequent principal component analysis showed that the samples no longer clustered by dataset (Fig. S7).

## Identifying differentially methylated probes and genes

We identified probes/genes that were methylated at significantly different levels between each tumor type and the batch-adjusted normal data. We used M-values and the limma package and included sex as a covariate. Limma uses the Benjamini-Hochberg False Discovery Rate method to adjust $p$-values (*Benjamini & Hochberg, 1995*).

When comparing methylation levels across the tumor types, we again used the limma package and included sex as a covariate. We did a pairwise comparison between

each pair of tumor types and used an adjusted $p$-value threshold of 0.05. We limited the results to probes with an absolute log fold-change threshold of 2 or greater for any pair.

### Gene-set analyses

We evaluated whether the highest-variance probes were significantly associated with known gene sets. To perform this analysis, we used the missMethyl package (*Phipson, Maksimovic & Oshlack, 2016*), which is part of the Bioconductor system. We used the same package to evaluate probes that were differentially methylated between each tumor type and the normal data.

### Inferring hypermethylation and hypomethylation states

To determine hypermethylation and hypomethylation status, we identified thresholds representing three standard deviations above or below the mean for each gene in the normal data. Then for each tumor sample, we compared the value for a given gene against the thresholds from the normal data. If a tumor's value was more than three standard deviations above the normal mean, we classified that tumor as being hypermethylated for that gene. If a tumor's value was more than three standard deviations below the mean of the normal samples, we classified that tumor as being hypomethylated for that gene.

### Mutual exclusivity

We evaluated whether aberrant methylation—either hyper- or hypomethylation—was mutually exclusive with somatic-mutation events for a given gene. We calculated the total number of times a gene was both mutated and aberrantly methylated in the same tumor. Next we permuted the methylation status for all tumors and kept mutation status constant. We repeated the permutation process 10,000 times to create an empirical null distribution to use as a baseline. For each permutation, we calculated the number of times a gene was both mutated and aberrantly methylated, then compared the number of times mutations co-occurred with aberrant methylation in the non-permuted data relative to the permuted data; finally, we calculated an empirical $p$-value based on these numbers.

### Oncogenes *vs* tumor suppressor genes

We identified genes known to be oncogenes or tumor suppressor genes, based on information from The Cancer Gene Census (*Tate et al., 2019*). We included only tier 1 genes classified as "oncogene" or "TSG" and ignored any for which it was ambiguous or unknown. To be classified as a tier 1 gene, there must be scientific evidence that a gene plays a role in cancer development and that mutations in the gene modify the activity of the associated protein. Several genes were listed in the Cancer Genome Census as both oncogenes and tumor suppressor genes; because of this ambiguity, we excluded these from our analysis.

## ABBREVIATIONS

**WT**      Wilms tumor
**CCSK**    Clear cell sarcoma of the kidney

| RT | Rhabdoid tumor |
| NBL | Neuroblastoma |
| OS | Osteosarcoma |
| TSG | Tumor suppressor gene |

## ACKNOWLEDGEMENTS

We thank the research participants who donated samples for molecular profiling and the researchers who consented to have their data shared publicly. The results published here are in whole or in part based upon data generated by the Therapeutically Applicable Research to Generate Effective Treatments (TARGET) initiative, phs000218, managed by the NCI. The data used for this analysis are available from the National Cancer Institute Genomic Data Commons (https://gdc.cancer.gov). Information about TARGET can be found at http://ocg.cancer.gov/programs/target.

### Funding

This work was funded by the College of Life Sciences at Brigham Young University for providing funding to Alyssa C Parker through a College Undergraduate Research Award. The funders had no role in study design, data collection and analysis, decision to publish, or preparation of the manuscript.

### Grant Disclosures

The following grant information was disclosed by the authors:
College of Life Sciences at Brigham Young University.

### Competing Interests

Stephen R. Piccolo is an Academic Editor for PeerJ.

### Author Contributions

- Alyssa C. Parker conceived and designed the experiments, performed the experiments, analyzed the data, prepared figures and/or tables, authored or reviewed drafts of the article, and approved the final draft.
- Badí I. Quinteros conceived and designed the experiments, performed the experiments, analyzed the data, prepared figures and/or tables, authored or reviewed drafts of the article, and approved the final draft.
- Stephen R. Piccolo conceived and designed the experiments, performed the experiments, analyzed the data, prepared figures and/or tables, authored or reviewed drafts of the article, and approved the final draft.

### Data Availability

All of the code used to perform the analysis is publicly available on Open Science Framework so that others can verify and build upon our work: Parker, Alyssa, and Stephen Piccolo. 2022. "Pediatric Cancer DNA Methylation." OSF. April 8. DOI 10.17605/OSF.IO/79YFB.

## Supplemental Information

Supplemental information for this article can be found online at http://dx.doi.org/10.7717/peerj.13516#supplemental-information.

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
