# Peer review of "The DNA methylation landscape of five pediatric-tumor types"

_PeerJ, doi:10.7717/peerj.13516_

## Round 0.1 · original submission · Major Revisions

Thank you for your submission. While the reviewers found the manuscript to be clear and well constructed, there were a small number of concerns around the experimental design. Please try and address these concerns as best as possible.

Reviewer 1 ·

Basic reporting

This is an interesting study that worth publication. The writing in clear, logical and easy to understand. Research questions were well defined. Methods and data descriptions had sufficient details. Overall, the analyses are technically and statistically sound.

Experimental design

Major concerns:

1) The authors calculated the average methylation for each gene (300 bp around TSS) first, then did the differential analysis. It makes more sense to do differential analysis at probe/CpG level first, then mapped the differential CpGs to genes.
2) During differential analyses (for example, comparing tumor to normal, or comparing different tumors), how the author handles confounding factors such as gender?

Validity of the findings

This is a meta-analysis project. All analyzed data (accession numbers) were provided. Conclusion are well stated.

Additional comments

Other minor problems:

• Line 9 -10: “Methylation is an epigenetic process of regulating gene expression in which methyl groups are attached to DNA molecules, often in promoter regions”. Such definition of “DNA methylation” is not accurate, and have these problems: 1) in the context of this manuscript, the audience know “Methylation” was referring to “DNA methylation”(5mC), but there are other type of methylations including RNA methylation (m6A) and histone methylation. 2) the methyl groups are attached to the 5th C of the cytosine ring. In somatic cells, 5mC most occurs in the CpG dinucleotides, and CpG CpG dinucleotides form CpG islands (CGIs). Many CGIs are located in promoters, but not all of them.
• Line 99-100: “We anticipated that many genes (such as tumor suppressor genes) would have consistently low methylation levels because those genes must be consistently expressed to properly regulate cellular division, growth, and other proliferation activities”. This is not always true, TP53 is a well-known tumor suppressor gene, but this gene is inactive or expressed at very low level in normal conditions.
• Line 238-240: “Because oncogenes are typically expressed at relatively high levels and tumor suppressor genes are typically expressed at relatively low levels, we expected that oncogenes would have higher methylation levels than tumor suppressor genes.” I don’t’ understand the logic here, if you assume oncogenes are “HIGH” expressed, you should expect “LOW” methylation at the promoter.
• Line 61: gene symbols should be in italics

Reviewer 2 ·

Basic reporting

This paper describes the process of collecting methylation data related to pediatric tumors and a set of normal samples. These samples were then analyzed to attempt to find genes which varied in methylation between the tumor and normal samples. The authors have also deposited all of their source code and the code appears to be well written and annotated. The paper is also well written and clear.

For the data presented in figure one, I’m wondering what the overall distribution of the gene-level methylation value looks like. I think you have two plotting options here, assuming you think this is a good recommendation. Option one would be two histograms showing the average methylation and another showing the CV values. Your other option would be a scatter plot of the average value vs CV (although you will probably have to use geom_hex with ~20000 data points). It would also be helpful to add a few lines to these plots to show where the categorization thresholds are.

Minor:

Typo line 38: “one type of genes” -> “one type of gene”
Typo line 40: “type of genes” -> “type of gene”

Experimental design

I suppose my largest problem with the entire design of this study is I don’t really think the samples you’ve gathered can be compared. Why would methylation data from fetuses through to 18 year olds from diverse tissues be comparable to unmatched tumor samples? I of course realize that there probably isn’t an ironclad great answer to this and you did somewhat address this by indicating that there weren’t dramatic differences in baseline data sets, but I’d like to see this examined more. Are all the genes in the high variance categories in Figure 1 showing significant differences between the normal data sets?

I’m also worried about using batch correction with the study ID as the batch correction factor. As far as I understand, the batch correction makes assumptions about the structure of the data that are only met when you have genuine repeats of an experimental setup, which you expect to show the same result. I think the diversity of these studies breaks that assumption. What does the methylation variance picture look like in the absence of batch correction? Specifically, if you do an analysis that is comparable to Figure 2 between the normal data sets, how many differences are you seeing?

If the other reviewer or editor don’t have a problem with combining these data sets without a deeper dive into differences in the normal data sets, then I’ll defer to their opinions.

Validity of the findings

No additional comments beyond those made in the experimental design section.

---

## Round 0.2 · accepted · Accept

Thank you for addressing the reviewer concerns in your revision of this interesting work. Congratulations again and thanks for your patience.

Reviewer 2 ·

Basic reporting

The authors have responded to all of my concerns and I see no problem with publication. I'll second the cover letter comment that negative results should be a welcome part of the publication record. Sometimes it's more interesting to see that an idea that should work doesn't in the context of a given paper, but can hopefully inspire others to see if it can in new situations.

Experimental design

No comment

Validity of the findings

no comment

Additional comments

No comment

---

## Author Rebuttal · Round 0.2

DEPARTMENT OF BIOLOGY

[Figure]

Dear Editors:

Thank you for reviewing our manuscript entitled, "*The DNA methylation landscape of five pediatric-tumor types*" (#2021:08:64894). We have carefully addressed both of the reviewers' comments and suggestions. In particular, Reviewer 2 expressed concern about using the "study ID as the batch correction factor." After considering this comment carefully, we realized that our original approach was problematic, in part because it would reduce the signals we were trying to find. In our original approach, we applied a batch-correction method across all datasets (normal and tumor), adjusting for differences between studies. However, in our analysis stage, we were looking for differences between datasets (either between normal samples and tumors or among tumor types). For the updated submission, we have modified our data processing and analysis pipeline and recomputed all of the results. We still do batch correction but only on the normal data (additional details below). Beyond this, we have added analyses that the reviewer requested regarding batch correction within the normal datasets. Our overall results have changed considerably—especially the genes that we identified as most relevant—but our overall message is largely the same. As with our previous submission, some of the results that we report are negative findings; however, we feel that reporting negative findings is integral to sound scientific practice, and we are confident that our manuscript meets PeerJ's editorial criteria.

We have also made edits to the manuscript to improve clarity.

Below we provide a detailed response to the editor's and reviewers' comments.

Warm regards,

Stephen R. Piccolo, PhD
Associate Professor
Department of Biology
Brigham Young University
(801) 422-7116
Stephen_Piccolo@byu.edu
* * *
**We have listed the editor's and reviewers' comments in gray text below. Our comments are in black, bolded text.**

Editor comments (Shawn Gomez)
MAJOR REVISIONS
Thank you for your submission. While the reviewers found the manuscript to be clear and well constructed, there were a small number of concerns around the experimental design. Please try and address these concerns as best as possible.

**Thank you.**

Reviewer 1 (Anonymous)
Basic reporting
This is an interesting study that is worth publication. The writing is clear, logical and easy to understand. Research questions were well defined. Methods and data descriptions had sufficient details. Overall, the analyses are technically and statistically sound.
Experimental design
Major concerns:

**Thank you for these positive comments.**

1) The authors calculated the average methylation for each gene (300 bp around TSS) first, then did the differential analysis. It makes more sense to do differential analysis at probe/CpG level first, then mapped the differential CpGs to genes.

**Thank you for this suggestion. We have reworked the analysis code so that the differential-methylation analysis is performed at the probe level first, then at the gene level. Additionally, we added probe-filtering steps that we had not included in the original manuscript. Furthermore, we reworked the way we adjusted for differences between datasets (in response to Reviewer 2). We believe these steps have led to a cleaner set of results.**

2) During differential analyses (for example, comparing tumor to normal, or comparing different tumors), how the author handles confounding factors such as gender?

**In our original submission, we had not been accounting for sex as a covariate. In response to your comment, we found that all of our datasets provide sex information along with the methylation data. We have modified our differential-methylation analyses so that sex is used as a covariate. (Sex was the only covariate for which data were available across all of the datasets.)**

Validity of the findings
This is a meta-analysis project. All analyzed data (accession numbers) were provided. Conclusions are well stated.

**Thank you.**

Additional comments
Other minor problems:

• Line 9 -10: "Methylation is an epigenetic process of regulating gene expression in which methyl groups are attached to DNA molecules, often in promoter regions". Such definition of "DNA methylation" is not accurate, and have these problems: 1) in the context of this manuscript, the audience know "Methylation" was referring to "DNA methylation"(5mC), but there are other type of methylations including RNA methylation (m6A) and histone methylation. 2) the methyl groups are attached to the 5th C of the cytosine ring. In somatic cells, 5mC most occurs in the CpG dinucleotides, and CpG CpG dinucleotides form CpG islands (CGIs). Many CGIs are located in promoters, but not all of them.

**Thank you for letting us know of the need to clarify this point. We have modified this part of the paper to say the following: "In one epigenetic process of regulating gene expression, methyl groups are attached at the 5-carbon of the cytosine ring, leading to 5-methylcytosine (5mC). In somatic cells, 5mC occurs mostly in CpG islands, which are often within promoter regions."**

• Line 99-100: "We anticipated that many genes (such as tumor suppressor genes) would have consistently low methylation levels because those genes must be consistently expressed to properly regulate cellular division, growth, and other proliferation activities". This is not always true, TP53 is a well-known tumor suppressor gene, but this gene is inactive or expressed at very low level in normal conditions.

**We changed this sentence to say, "We anticipated that many genes (including some but not all tumor suppressor genes) would have relatively low methylation levels because those genes often regulate cellular division, growth, and other proliferation activities." We also updated the related part about oncogenes.**

• Line 238-240: "Because oncogenes are typically expressed at relatively high levels and tumor suppressor genes are typically expressed at relatively low levels, we expected that oncogenes would have higher methylation levels than tumor suppressor genes." I don't understand the logic here, if you assume oncogenes are "HIGH" expressed, you should expect "LOW" methylation at the promoter.

**Thank you for pointing out this discrepancy. We have addressed it. This section now reads: "Because oncogenes are typically expressed at relatively low levels under normal conditions and tumor suppressor genes are typically expressed at relatively high levels, we expected that oncogenes would have higher methylation levels than tumor suppressor genes."**

• Line 61: gene symbols should be in italics

**We have added italics for these genes. Thank you for your careful and fast review!**

Reviewer 2 (Anonymous)
Basic reporting
This paper describes the process of collecting methylation data related to pediatric tumors and a set of normal samples. These samples were then analyzed to attempt to find genes which varied in methylation between the tumor and normal samples. The authors have also deposited all of their source code and the code appears to be well written and annotated. The paper is also well written and clear.

**Thank you for these positive comments.**

For the data presented in figure one, I'm wondering what the overall distribution of the gene-level methylation value looks like. I think you have two plotting options here, assuming you think this is a good recommendation. Option one would be two histograms showing the average methylation and another showing the CV values. Your other option would be a scatter plot of the average value vs CV (although you will probably have to use geom_hex with ~20000 data points). It would also be helpful to add a few lines to these plots to show where the categorization thresholds are.

**Thank you for this suggestion. We have added histograms showing the distribution of methylation values at the probe level and gene level, respectively. Additionally, we have added scatter plots**

**showing the relationship between median beta value and CV at the probe level and gene level, respectively.**.

Minor:

Typo line 38: "one type of genes" -> "one type of gene"
Typo line 40: "type of genes" -> "type of gene"

**Thank you. We have adjusted this wording.**

Experimental design
I suppose my largest problem with the entire design of this study is I don't really think the samples you've gathered can be compared. Why would methylation data from fetuses through to 18 year olds from diverse tissues be comparable to unmatched tumor samples? I of course realize that there probably isn't an ironclad great answer to this and you did somewhat address this by indicating that there weren't dramatic differences in baseline data sets, but I'd like to see this examined more. Are all the genes in the high variance categories in Figure 1 showing significant differences between the normal data sets?

**Thank you for this comment. To respond to this question, we did two things. As we now describe in Methods, we performed some additional probe-filtering steps (based on recommendations from Alicia Oshlack's group). This removed many high-variance (and likely problematic probes). In addition, we explored the normal data and added some results to the manuscript. We evaluated genes that exhibited high variance across all normal datasets and asked whether that variance was due to high variation in general or whether that variance was caused by only one (or some) of the datasets. We added the following to the manuscript: "Of the 19,184 genes, 14,706 (76.7%) were consistent, 3,481 (18.1%) were semiconsistent, and 997 (5.2%) were inconsistent. These findings demonstrate that even though the normal datasets differed based on cell type and population ancestral groups, gene-level methylation levels were largely consistent under normal conditions. However, some patterns were specific to a given dataset. of the 2,106 genes that were highly variant for at least one dataset, 991 (47.1%) genes were highly variant when calculated across all the normal datasets, showing that genes with high variance in one dataset often did not exhibit high variance more generally." In addition, we modified Figure 1 to more clearly show the number of probes/genes that fell into each category.**

I'm also worried about using batch correction with the study ID as the batch correction factor. As far as I understand, the batch correction makes assumptions about the structure of the data that are only met when you have genuine repeats of an experimental setup, which you expect to show the same result. I think the diversity of these studies breaks that assumption. What does the methylation variance picture look like in the absence of batch correction? Specifically, if you do an analysis that is comparable to Figure 2 between the normal data sets, how many differences are you seeing?

If the other reviewer or editor don't have a problem with combining these data sets without a deeper dive into differences in the normal data sets, then I'll defer to their opinions.

**Thank you for mentioning this issue. After considering your comment, we realized that performing batch correction this way was problematic, in part because it would remove much of the signal we were attempting to find. In our analyses, we were looking for genes that were methylated differently between tumor and normal tissues, but by treating each dataset as a batch, much of the meaningful signal in the data was likely removed. We still think that batch adjustment is important for the normal**

datasets (to create a combined reference), but we have done this for the normal data only. In addition, we did pairwise comparisons between normal datasets (before batch correction) to identify probes that were differentially expressed in this context. So that our aggregated normal dataset would be a more reliable reference, we removed any probe from our analysis that showed significant differences between any pair of normal datasets (using the same filtering criteria that we used when comparing tumor vs. normal). 49 probes met this criterion.

Validity of the findings
No additional comments beyond those made in the experimental design section.

**Thank you for your careful and fast review!**